# Learning to map between ferns with differentiable binary embedding networks

**Max Blendowski**                                        BLENDOWSKI@IMI.UNI-LUEBECK.DE

**Mattias P. Heinrich**                                    HEINRICH@IMI.UNI-LUEBECK.DE

*Institute of Medical Informatics, Universität zu Lübeck, Lübeck, Germany*

## Abstract

Current deep learning methods are based on the repeated, expensive application of convolutions with parameter-intensive weight matrices. In this work, we present a novel concept that enables the application of differentiable random ferns in end-to-end networks. It can then be used as multiplication-free convolutional layer alternative in deep network architectures. Our experiments on the binary classification task of the TUPAC'16 challenge demonstrate improved results over the state-of-the-art binary XNOR net and only slightly worse performance than its 2x more parameter intensive floating point CNN counterpart.

**Keywords:** End-To-End Trainable Ferns, Network Efficiency, Binary Embedding.

## 1. Introduction

Nearly all current deep learning methods rely on a vast amount of floating point operations and rather simplistic combinations of trainable convolution filters and rectifying non-linearities. Another direction of machine learning that is based on non-linear multi-dimensional mapping through ensembles of binary decision trees (Criminisi et al., 2012) or random ferns (Ozuysal et al., 2009) has become less relevant due to the difficulty or inability to embed these approaches into end-to-end trainable networks. However, random ferns prove to serve as fast and efficient feature extractors in connection with separately trainable layers (Kim et al., 2019). Designing differentiable decision boundaries in deeper binary trees is considered a very challenging task therefore previous work has focused on combining CNNs with differentiable (neural) forests (Kontschieder et al., 2015). In parallel, much research work has recently been devoted to binary networks (Rastegari et al., 2016) that avoid memory- and computation-intensive floating point matrix multiplications. We propose a method to efficiently use random ferns within an end-to-end trainable architecture to replace convolutions without using floating point multiplications.

## 2. Method

To explain the proposed method, first, we will revisit the concept of a random ferns ensemble without optimisation. Next, our proposed differentiable binary embedding with weighted sums is described and extended to multi-layer and convolutional architectures.

For the most part, the procedure during a standard convolution is identical to our method (unfold $\doteq$ IM2COL (Chetlur et al., 2014), matrix-multiplication, fold), since only

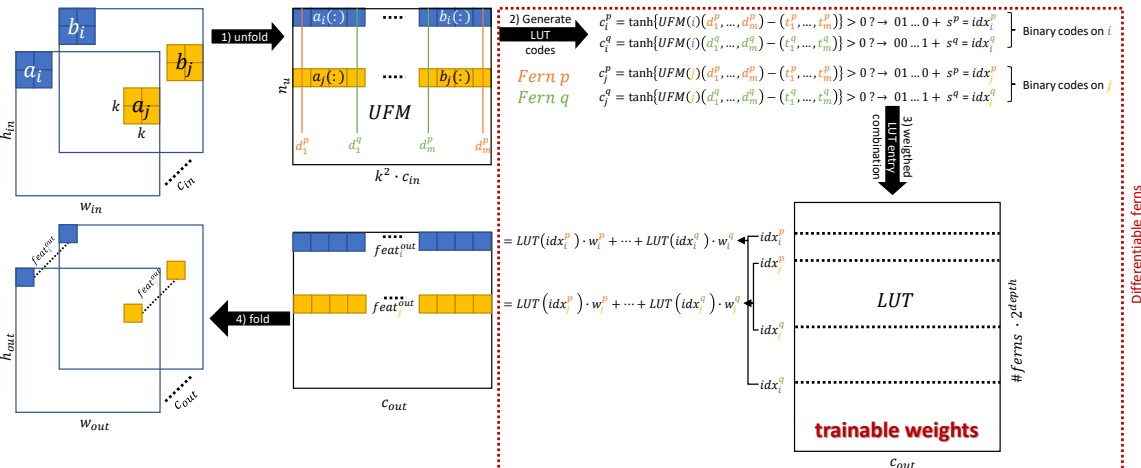

Figure 1: Convolution Drop-in Replacement: The unfolded-tensor-matrix · weight-matrix multiplication is replaced by a fern ensemble using an EmbeddingBag-layer as LUT. Measuring the proximity of continuous valued feature vectors to binary strings to compute a weighted sum of LUT entries allows differentiability.

the matrix-multiplication part is replaced by the differentiable random fern implementation. Therefore, our method is a potential drop-in replacement for convolutions using a Look-Up-Table (LUT) instead of multiplications.

To generate a classical random fern ensemble classifier, as a first step, for every fern $k$, corresponding to the depth $m$, two sets will be randomly drawn. The first set consists of a random subset of the input feature dimensions $(d_1^k, ..., d_m^k)$ and the second set contains a number of thresholds $(t_1^k, ..., t_m^k)$. In contrast to optimized decision trees, where different dimensions of a feature vector will be examined along its path to a leaf-node, traversing a fern yields always the same feature dimension sequence whose contents will be compared with the fixed thresholds (see orange and green lines in Figure 1 at the unfolded feature matrix (UFM) as fixed dimensions of interest for the Ferns $p$ and $q$). Every comparison results in a binary output and encodes as binary string an index to access the class specific histograms of each fern.

In contrast to previous work, the output of each fern is in our case not a data driven (normalised) histogram but directly learned as a feature vector. Inspired by Natural Language Processing (Mikolov et al., 2013), we use the EmbeddingBag-layer to map a dictionary (here of size $\#ferns \cdot 2^m$) into a different-dimensional output space - effectively implementing all different class histograms of a fern ensemble into a single large LUT. Inside the red-dotted box in Figure 1, we follow the original fern algorithm by feeding rows of the UFM (generated with the IM2COL operator) through the random fern ensemble - except for the minor deviation of applying a $tanh$ function after the threshold substraction, resulting in a vector $c_u^k$ per fern $k$ and row $u$. Taking the sign of $c_u^k$, converting it to its decimal representation and adding an appropriate offset $s^k$ per fern gives access to the according LUT index position $idx_u^k$ and its embedding weights.

The key observation to gain differentiability for these discretely addressed LUT embeddings is the fact that while the feature indices $idx_u^k$ based on the UFM provide no gradient to train networks, a scalar instance weight $w_u^k$ that measures the proximity of continuous valued feature vectors to binary strings is very suitable to enable end-to-end training. Here it is obtained by computing the mean distance of absolute $c_u^k$ entries to 1: $w_u^k = \|abs(c_u^k) - \mathbf{1}\|_2$. The weighted sums of these LUT entries form the unfolded output feature matrix containing $feat_u^{out}$, before a final COL2IM operation reshapes the data.

Table 1: Classification results for the Tumor Proliferation Assessment Challenge (TUPAC)

| Input Patches | Architecture | # Params | Energy consumption | Accuracy |
|---|---|---|---|---|
| | XNOR net | $\approx$ 80k | 2.45 $\mu J$ | 82.66% |
| | Vanilla net | $\approx$ 80k | 65.5 $\mu J$ | **84.23%** |
| | Fern net (ours) | $\approx$ **40k** | **1.01** $\mu J$ | 83.97% |

## 3. Experiments & Results

We perform our experimental validation on the binary classification task of the Tumor Proliferation Assessment Challenge 2016 and show that relatively shallow ferns as networks with very few trainable weights can be learned that enable high classification accuracy. As baseline comparisons, we use a Vanilla CNN architecture and its conversion as XNOR net (Rastegari et al., 2016).

In all 3 experiments, we use the same network architecture: a 5-layer network defined by the following encoding scheme $(c_{in}, c_{out}, kernelsize, stride, norm)$: 1) $(3, 64, 5, 2, BN)$, 2) $(64, 64, 3, 2, BN)$, 3) $(64, 64, 3, 2, BN)$, 4) AdaptiveAvgPool, 5) $(64, 2, 1, 1, -)$. The Vanilla net uses ReLU activation functions after the first 3 layers, whereas the XNOR counterpart achieves non-linearity already by its input binarization. While we change the backbone using our Fern-Ensemble layers, the spatial operations (unfolding & folding) remain untouched for the Fern net. In every layer, we use 24 ferns with a depth of only 3. Here, index-binarization provides the non-linearity. With only half the parameters, our approach falls just short of the Vanilla CNN implementation (Table 1) and outperforms the XNOR net. Regarding the energy comsumption of processing a single input image according to (Hubara et al., 2016), our Fern net is by far the most efficient approach.

## 4. Conclusion

We presented a novel approach that enables the use of *random ferns within an end-to-end trainable convolutional architecture* and demonstrates impressive classification results that are on-par with state-of-the-art binary XNOR nets and without using floating point multiplications. Spatial convolutions can be easily integrated into fern-like architectures by employing the IM2COL operator. In contrast to conventional ferns that build class histograms purely data-driven, we learn the embedding directly - following the end-to-end trainable paradigm of learning task specific feature extractors and classifiers simultaneously.

## Acknowledgments

This work was supported by the German Research Foundation (DFG) under grant number 320997906(HE 7364/2-1). We gratefully acknowledge the support of the NVIDIA Corporation with their GPU donations for this research.

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
