# OpenReview forum: "Learning to map between ferns with differentiable binary embedding networks"
_MIDL.io/2020/Conference — MIDL 2020_

### Official Review · AnonReviewer2 · 2020-03-09
**interesting topic but vague presentation**

**Rating:** 1
**Confidence:** 5

**Review:**

This paper relies on Fig. 1 to convey most parts of the important ideas. Unfortunately, the figure doesn't show all the key components clearly.

the main innovation seems to be replacing a multiplication operation with a lookup table + trainable weighted sums.  Since convolution can be implemented as im2col followed by a matrix multiplication, I feel it's more appropriate to claim it's a fast convolution by caching some binary encoded results.  It's not clear how much memory it takes and how the energy consumption calculated in terms of both memory access and arithmetic operations. The paper also claims "without using floating-point multiplications", but there're floating number weighted sums as shown in Fig. 1.

---

### Official Review · AnonReviewer1 · 2020-03-13
**Learning to map between ferns**

**Rating:** 3
**Confidence:** 4

**Review:**

The main contribution of the paper is a novel approach of taking advantage of ferns and achieving the (almost) same performance of a Vanilla Net at TUPAC challenge with only half of network size and 1/60th of energy consumption. Removal of floating point multiplications are attributed to using a Look-Up Table that holds the trainable weights, whose indices come from binary string comparison between feature vectors, hence introducing non-linearity to the system. Overall, the paper is concise and benefits great from the fact that the authors implemented their novel approach on TUPAC challenge, which is a publicly known/studied dataset contest.
The authors results are tested on a public challenge, and their findings on the public challenge validate their methodological improvements.
The authors reduce the net size substantially by removing multiplications with a Look-Up Table and improves the accuracy by learning the feature embeddings instead of using histogram. Their approach allows their implementation to achieve the (almost) same accuracy as Vanilla Net and outperform the XNOR net while having a substantially small network/parameter size.

Some implementation details were given without explanation. This could be attributed to the fact that the paper was submitted for a short paper track and did only have a 3-page limit. Nevertheless, the authors don't explain why they chose tanh for the threshold subtraction. They also don't explain the reasoning behind choosing ferns 3 for the depth of the ferns they use at every layer.

Spatial convolutions are integrated thanks to IM2COL operator. The authors might be valuable to explain the benefits that they got from that operator more to help the reader with her understanding.

The LUT size is #ferns * m. This LUT becomes a hash table where the lookup is in constant time, but the storage is still needed. The paper was unclear whether they included the size of the look-up table in their parameter calculation in Table 1 since building that LUT will still require space. One still might say choosing 24 ferns at every layer with a depth 3 will not consume substantial space for the generation of the lookup table, but greater number of layers and ferns might make this approach converge to the vanilla net in terms of size if one wants to scale it up.

Also is there a particular reason this method is used or benefits medical machine learning?

---

### Official Review · AnonReviewer4 · 2020-03-13
**Not so clear...**

**Rating:** 2
**Confidence:** 3

**Review:**

Summary: Authors propose to replace the matrix-multiplication part of a convolutional layer with a differentiable random fern (as defined in Özuysal et al. IEEE TPAMI 2009). It is shown that this method reduces by two the number of parameters, with respect to a standard CNN, and preserves almost the same performance in terms of classification accuracy.

Remarks:
1- the paper is quite difficult to read and understand. Many concepts such as ferns, IM2COL, UFM and EmbeddingBag-Layer are not sufficiently explained in the paper.

2- A fern, as introduced by Özuysal and colleagues, is a small set of binary tests that is used with a Semi-Naive Bayesian approach in a problem of classification. It's not clear how exactly ferns can replace matrix-multiplication. Authors should better explain this point.

3- Many choices are not well motivated or explained such as: the use of tanh, the offset s^k, the definition of w_u^k

4- Results seem interesting and it's a pity that the paper is not so clear. Authors probably need more space to better explain their algorithm and all related concepts.

---

### Official Review · AnonReviewer3 · 2020-03-19
**Differentiable ferns for end-to-end training of architectures**

**Rating:** 3
**Confidence:** 4

**Review:**

The paper proposes embedding ferns as an alternative to convolutional layers for deep learning architectures. It departs from standard random ferns and variants in that it is a drop-in replacement and allows for end-to-end training of architectures.

The abstract is well structured and relatively easy to follow, and overall it can be interesting for MIDL.

What is gained from moving from a convolutional layer to a fern? Ultimately it looks like the computational complexity of the proposed layer, and the memory footprint is going to be similar to that of a standard convolutional layer. There are $c_{out}\times \text{#ferns} \times 2^{depth}$ trainable parameters (plus the fixed ones) where a convolutional layer would have at worst, $c_{out}\times c_{in}\times k^2$.
It is not clear whether there are settings of the depth and number of ferns such that the number of parameters, computational complexity and/or memory usage will be reduced by an order of magnitude at little cost in performance, since even a depth of $3$ leads to similar number of parameters as a $3\times 3$ kernel.

In terms of operations, floating point multiplications from convolutions are replaced by $\Vert \, \cdot \Vert^2$, which expands one way or the other to similar floating point multiplication or squaring, plus the $tanh$.

What is the energy consumption gain attributable to? Also why not report other metrics that relate to computational/memory complexity?

---

### Meta-Review · Area_Chair1 · 2020-03-27
**MetaReview of Paper133 by AreaChair1**

**Rating:** 3

**Metareview:**

The majority of reviewers acknowledge that the idea of using random ferns as a replacement for convolutions is worthwhile to investigate. There are some concerns regarding the presentation and the lack of clarity where gains in energy consumption come from. Overall, it seems the methodological idea is interesting and suitable to be discussed at MIDL 2020.

**Paper Type:**

methodological development

---

### Decision · Program_Chairs · 2020-04-11

Accept